# Evaluating the Efficacy of the Male Annihilation Technique in Managing Oriental Fruit Fly (Diptera: Tephritidae) Populations through Microscopic Assessment of Female Spermathecae

**DOI:** 10.3390/insects15100796

**Published:** 2024-10-14

**Authors:** Dian Zhou, Meizhu Liu, Jing Wang, Fang Fang, Zhanbin Gong, Daihong Yu, Yunguo Li, Chun Xiao

**Affiliations:** 1Faculty of Plant Protection, Yunnan Agricultural University, Kunming 650201, China; zhoudian_hn@163.com (D.Z.); qingxinzr0694@163.com (M.L.); xiaojingxwj@163.com (J.W.); fangf3264@163.com (F.F.); 2Apple Industry Development Center, Zhaotong 657099, China; gongzb2024@163.com; 3Plant Protection and Quarantine Station, Yuanjiang County, Yuxi 653300, China; yudaihongyu@163.com

**Keywords:** *Bactrocera dorsalis*, male annihilation technique, proportion of mated females, spermatheca

## Abstract

**Simple Summary:**

*Bactrocera dorsalis* (Hendel) is a notorious horticultural pest. The male annihilation technique (MAT) is a key strategy in the pest management program of *B. dorsalis*. However, there is currently no method to monitor the effectiveness of MAT in real time. In this study, it was discovered under a microscope that sperms can be detected in the spermathecae and the ventral receptacle of mated *B. dorsalis* females. Furthermore, sperms are more easily observed in the spermathecae and can be retained there for more than 50 days. Field investigations demonstrated that the proportion of mated females was less than 81.2% in an abandoned mango orchard, while it was reduced to less than 36.4% in another mango orchard where MAT was applied. This indicates that dissecting the spermathecae to observe the presence of sperms is a reliable method to accurately monitor the proportion of mated *B. dorsalis* females in the fields. This method can be used in real time and accurately assess the control efficiency of MAT in *B. dorsalis* management.

**Abstract:**

The male annihilation technique (MAT) plays a crucial role in the pest management program of the oriental fruit fly, *Bactrocera dorsalis* (Hendel) (Diptera: Tephritidae). However, a suitable method for real-time and accurate assessment of MAT’s control efficiency has not been established. Laboratory investigations found that motile sperms can be observed clearly under the microscope when the spermathecae dissected from mated females were torn, and no sperms were found in the spermathecae of virgin females. Furthermore, it was confirmed that sperms can be preserved in the spermathecae for more than 50 days once females have mated. Laboratory results also indicated that proportion of mated females decreased from 100% to 2% when the sex ratio (♀:♂) was increased from 1:1 to 100:1. Further observation revealed that there were no significant differences in the superficial area of the ovary or spermatheca between mated females and virgin females. Field investigations revealed that the proportion of mated females (PMF) could reach 81.2% in abandoned mango orchards, whereas the PMF was less than 36.4% in mango orchards where MAT was applied. This indicates that the PMF of the field population can be determined by examining the presence of sperms in the spermathecae. Therefore, we suggest that this method can be used to monitor the control efficiency when MAT is used in the field.

## 1. Introduction

The oriental fruit fly, *Bactrocera dorsalis* (Hendel) (Diptera: Tephritidae), severely damages more than 300 types of commercial or edible fruits [1,2]. Mated females lay eggs in ripe fruits, and the larvae feed on the fruits and pupate in the soil [3]. Therefore, the adult stage is the only exposed stage of the life cycle, and the adult lure technique is the most effective and environment-friendly control method [4]. Methyl eugenol (ME) [1,2-dimethoxy-4-(2-propenyl)benzene], an aromatic compound from various plants, has been confirmed to have a strong attraction and feeding responses in sexually mature males [5,6,7]. The extensive use of ME-based traps in the field to attract and kill male flies is referred to as the male annihilation technique (MAT) [8,9]. Field practice has demonstrated that male flies can be eradicated or suppressed to low numbers using certain densities of ME traps in managed areas [10].

Fruit infestation rates [11] and population density [12] are recognizable evaluation parameters of control efficiency. For *B. dorsalis*, the evaluation of MAT involves assessing the proportion of mated females (PMF) and fruit infestation rates [13,14]. Fruit infestation rate directly reflects the actual damage caused by pests to the host and serves as an ultimate indicator for evaluating control efficacy. Thus, although fruit infestation rates are easily obtained, the data do not reflect a decrease in the number of mated females in real time. Moreover, real-time data on PMF are vital in adjusting ME trap density in the field. Therefore, rapid and accurate investigations of the proportion of mated *B. dorsalis* females are crucial in the evaluation of MAT performance in the field.

Common methods for determining whether a female insect has mated include inspection of hatching rate [15,16], spermatophore [17,18], morphological changes [19], and sperm in their spermathecae [20]. As *B. dorsalis* females lay eggs in the pulp of fruit, the collection and observation of hatched eggs in the field are challenging. Additionally, tephritid male flies do not produce spermatophores during mating [21]. Thus, the establishment of another approach is necessary to determine mated females.

For insects, seminal proteins from males can promote the ovarian development of females [22,23]. In female fruit flies, mating behavior occurs only when the ovaries are fully mature [21], and the stages of sexual development are categorized based on the characteristics of ovarian development [24,25]. An insect research report states that the superficial area of the spermathecae significantly changes in mated females compared with virgin females [26]. Therefore, observing the ovaries and spermathecae of mated females is necessary to determine whether the characteristics of their development reflect the mating status of *B. dorsalis* females.

In mated tephritid females, sperm can be detected in the spermathecae and ventral receptacle [20,27,28]. Although the ventral receptacle is an organ for short-term sperm storage, field research for sterile insect technique has indicated that in the *Ceratitis capitata* (Wiedemann) and *Anastrepha ludens* (Loew), dissecting the ventral receptacle can effectively prevent false-negative judgments of female mating status based on spermathecae dissection alone [29,30]. Therefore, determining the mating status of female *B. dorsalis* requires dissecting not only the spermathecae but also the ventral receptacle.

The life cycle of *B. dorsalis* spans more than 3 months, and females lay eggs multiple times [31]. The females typically mate only once in their entire life cycle [32,33]. Therefore, determining the duration of sperm preservation in *B. dorsalis* females and verifying whether mated females have depleted all sperms through oviposition (i.e., the existence of sperm-depleted mated females in the field) are necessary to determine whether the sperm detection method is suitable for field evaluation of PMF.

In summary, this research article reports a rapid method to detect the PMF of *B. dorsalis*. We assessed the mating status of females by dissecting their ovaries, spermathecae, and ventral receptacle, as well as evaluating the long-term sperm storage ability of the spermathecae. Additionally, we simulated the population sex ratio (females:males) imbalance caused by using MAT and evaluated the changes in PMF by assessing female mating status. Furthermore, we investigated the PMF in an orchard where MAT was implemented and in an abandoned orchard.

## 2. Materials and Methods

### 2.1. Insects

Mangoes infested by *B. dorsalis* were collected and transferred to the laboratory from an abandoned orchard in Yuanjiang, Yuxi, Yunnan, China. The environmental conditions were maintained at a temperature of 25 ± 1 °C, relative humidity of 60 ± 10%, and a photoperiod of 14 L:10 D [34]. These mangoes were placed in screened containers with sand at the bottom for four weeks, by which time all larvae had exited the fruit, dropped into the sand, and pupated [13]. After emergence, the adults were transferred to rearing cages (40 cm × 40 cm × 40 cm, approximately 500 flies per cage) and provided with food (10% yeast extract, 30% white sugar, and 1% agar) and water [35]. Once mating behavior began, an oviposition bottle was provided as the oviposition site for mated females [36]. Their eggs were collected daily and transferred to rearing vessels (1 L) with a larval diet (10% yeast extract, 30% white sugar, 15% wheat bran, 1% agar, and 44% water). The vessels were placed on fine sand where the larvae pupated. Pupae were regularly collected and placed in rearing cages. After adult emergence, some females and males were mixed at a sex ratio of 1:1 (♀:♂), whereas others were reared separately. Wild flies were annually collected from the orchard and crossed with the laboratory population to maintain genetic diversity. At the time of the experiment, the females were 9–10 days old and presumed sexually mature (more than 95% of females in mixed cages were mated by 7 days [37]).

### 2.2. Experiment 1: Effect of Mating Status on the Area of the Ovaries and Spermathecae

Once the mating behavior began, the flies (9 days old) were gently transferred to a new rearing cage. The mated females (10 days old) were placed in PBS buffer (phosphate-buffered saline [38], 137 mM NaCl, 2.7 mM KCl, 4.3 mM Na_2_HPO_4_, and 1.4 mM KH_2_PO_4_, pH 7.3). Their ovaries and spermathecae (Figure 1) were removed under a stereomicroscope (Nikon, SMZ445, Tokyo, Japan). Photographs were captured at 80× magnification to calculate the superficial area of the ovary and spermatheca. The experiment was repeated 50 times to compare the mated females and virgin females (10 days old).

### 2.3. Experiment 2: Observation of Sperms

The mated females (10 days old) were dissected under a stereomicroscope, and the spermathecae and ventral receptacle (Figure 1) were removed and placed in PBS on a slide. The spermathecal wall and the ventral receptacle were torn using forceps to check for the presence of sperms. The mated females were confirmed when motile sperms were observed under a microscope (Motic, Stellar 1-T, Xiamen, China). Then, the slide was placed in a shaded, well-ventilated area to air-dry and dyed with acid fuchsin (Solarbio, Beijing, China) for approximately 5 s. Each slide was then observed and photographed under a microscope. The experiment was repeated 100 times to compare the mated females and virgin females (10 days old).

### 2.4. Experiment 3: Sperm Retention Duration

Ten-day-old females that had just mated for the first time were divided into two groups: some were placed in a breeding cage with food and an oviposition bottle, whereas others were placed in a cage with food but no oviposition bottle. After 2, 4, 6, 8, 10, 20, 30, 40, and 50 days, ten females were removed from each cage and dissected under a stereomicroscope to check for the presence of sperms in the spermathecae. This experiment was repeated three times.

### 2.5. Experiment 4: Proportion of Mated Females with Varying Sex Ratios

Unmated 9-day-old flies were placed in cages at the sex ratios of 1:1, 2:1, 4:1, 5:1, 20:3, 10:1, 20:1, and 100:1 (females:males) with 100 females, and the respective number of males in each cage. After 48 h, all females’ spermathecae were dissected to determine the mating status, then the PMF at different sex ratios was calculated. The experiment was repeated five times.

### 2.6. Experiment 5: Field Evaluation of the Proportion of Mated Females

The investigation was conducted at two mango orchards of Yuanjiang, Yuxi, Yunnan, China (Figure 2). One site was an abandoned mango orchard A (Tainung No. 1, 101°58′10″ E, 23°36′03″ N, 17 hm^2^), with plant spacing of 2–5 m, plant height of greater than 3 m, and no field sanitation or management. The other site was mango orchard B (Tainung No. 1, 101°57′33″ E, 23°35′48″ N, 18 hm^2^) undergoing MAT, with plant spacing of 5–7 m, plant height of approximately 2 m, and cursory field sanitation. The two orchards were approximately 700 m apart. Orchard B underwent 4 months of MAT from 1 May to 1 September of each year. ME traps (LC Tech Co., Ltd., Kunming, China) were placed at intervals of approximately 15 m. Each lure (plastic ampoules) contained 2 mL of ME (≥98%, TCI, Tokyo, Japan) and was changed monthly. The investigations were performed on 1 and 8 June 2022, and 1 and 9 June 2023. At 8 AM, 10 similarly sized ripe fruits (collected under the mango trees, totaling approximately 600 g) were piled together, and a sticky board was fixed 6–10 cm above the fruits to capture females (Figure 3). At approximately 7 PM, three sticky boards were collected and brought to the laboratory to check the intraday PMF.

### 2.7. Statistical Analyses

All statistical analyses were conducted in RStudio v. 4.3.3 [39]. The marginal means were estimated using the “emmeans” package and its corresponding function [40].

Images of the ovary and spermatheca with corresponding scales were loaded into the ImageJ software (available at http://imagej.nih.gov/ij/, accessed on 21 April 2022) to extract their superficial areas [26]. Independent samples t-tests were conducted on the area data of mated and virgin females.

As the number of females in cages with different sex ratios was the same (*n* = 100), generalized linear models were used to assess the effects of sex ratios on PMF, and the PMF was translated into the number of mated females (negative binomial distribution). Because of zero variance in random effects, generalized linear models were used to analyze the effects of MAT or sampling time on PMF in orchards, and the PMF was translated into numbers of mated and virgin females (binomial distribution). The “glm” or “glmer” function in the “lme4” package [41] was used to build models. Tukey’s honestly significant difference test was used for pairwise post hoc comparisons.

## 3. Results

### 3.1. Experiment 1: Effect of Mating Status on the Area of the Ovaries and Spermathecae

The ovarian superficial area of mated females (2.57 ± 0.07 mm^2^) exhibited no significant difference compared with that of virgin females (2.72 ± 0.06 mm^2^) (t = 1.713, df = 98, *p* = 0.090) (Figure 4a). No significant difference was found in the spermathecal superficial area between mated females (0.0201 ± 0.0005 mm^2^) and virgin females (0.0189 ± 0.0004 mm^2^) (t = 1.863, df = 98, *p* = 0.065) (Figure 4b).

### 3.2. Experiment 2: Observation of Sperms

When the spermathecae of mated females were torn, the sperms swimming out of the spermathecae were observed (Figure 5a). In contrast, the spermathecae of virgin females were empty (Figure 5b). Air-dried slides revealed sperm mass and sperms near the spermatheca (Figure 6). However, the head of the sperms was not observable. Sperms within the alveolus of the ventral receptacle of mated females were difficult to observe (Figure 7b). However, sperms were visible on all slides created from the spermathecae of mated females.

### 3.3. Experiment 3: Sperm Retention Duration

Sperm was detected in all dissected spermathecae of females irrespective of the time elapsed since first mating or whether the female was provided an oviposition bottle.

### 3.4. Experiment 4: Proportion of Mated Females with Varying Sex Ratios

The sex ratio significantly affected the PMF (χ^2^ = 70.74, df = 1, *p* < 0.001) (Figure 8). As the sex ratio became more female-biased, the proportion of females with sperm in their spermathecae gradually decreased. At a sex ratio of 1:1, all females had mated. At a sex ratio of 100:1, PMF was only 2.0% ± 0.3%.

### 3.5. Experiment 5: Field Evaluation of the Proportion of Mated Females

MAT significantly influenced the PMF in mango orchards, as evidenced by the results for 2022 (χ^2^ = 36.52, df = 1, *p* < 0.001, Figure 9a) and 2023 (χ^2^ = 51.15, df = 1, *p* < 0.001, Figure 9b). In 2022, during the same sampling times, Orchard A had significantly higher PMF than Orchard B, with rates on 1 June at 81.20 ± 5.26% for Orchard A and 32.34 ± 1.34% for Orchard B (*p* < 0.001) and on June 9 at 79.53% ± 11.56% for Orchard A and 36.39 ± 1.06% for Orchard B (*p* < 0.001). In 2023, at the same sampling times, the PMF of Orchard A was significantly higher than that of Orchard B (1 June, 73.15 ± 3.34% and 30.07 ± 14.05%, *p* < 0.001; 9 June, 76.09 ± 4.02% and 15.32 ± 2.15%, *p* < 0.001). Sampling time did not significantly influence the PMF (2022, χ^2^ = 0.01, df = 1, *p* = 0.933; 2023, χ^2^ = 0.22, df = 1, *p* = 0.642), indicating no significant differences between two investigations in the same year at the same orchards (Orchard A 2022, *p* = 0.653; Orchard B 2022, *p* = 0.662; Orchard A 2023, *p* = 0.777; Orchard B 2023, *p* = 0.275).

## 4. Discussion

Although the control efficacy of MAT can be evaluated with fruit sampling, the relevance of this study is that having an accurate method to determine a reduction in the PMF helps better understand the reasons for a low or no drop in fruit infestation levels.

Generally, *B. dorsalis* does not damage the fruit before ripening [42,43]. Therefore, the application of MAT before fruit ripening will achieve the best control efficiency. However, it is challenging to evaluate the control efficacy of MAT at the unripe stage because of the absence of effective female lure techniques and reliable evaluation methods. Although there have been recent advancements in female lures of *B. dorsalis* [44,45], the actual effectiveness in the field has not fulfilled expectations. Consequently, we used the ripe fruits plus sticky boards to trap female flies in our field investigation, and this approach adequately met the demands of investigations according to practical outcomes (Figure 3).

The reproductive tract structure of *B. dorsalis* (Figure 1) is similar to that of *C. capitata* [29]. The sperm structure we observed (Figure 5a and Figure 6) is very similar to that described in *C. capitata* [46]. For *C. capitata*, sperms are typically observed by simply squashing the spermathecae on a slide to release the sperms [29]. However, in *B. dorsalis*, we found that the ruptures of the spermathecal wall were often obstructed by a gelatinous matrix, which blocked the release of sperms (Figure 7a). This made the squashing method less effective for observing sperms in *B. dorsalis*. Therefore, our findings indicate that directly tearing the spermatheca, rather than squashing it, is a more effective technique for releasing and observing sperms in this species.

The laboratory results indicated that sperms were present in the spermathecae for more than 50 days. Under natural conditions, *B. dorsalis* females typically reproduce one generation in approximately 1 month [47], and females are generally monandrous [32]. Thus, monthly or more frequent investigations using our dissection method can reliably determine the PMF. The results of the field investigations indicated that examining the sperms in the spermathecae of female flies to assess mating status is practical (Figure 9). The detection of sperms within female flies accurately assesses their mating status, which adequately meets the requirements of field investigations. Therefore, it is proposed that our method be used in future field studies to evaluate the control efficiency of MAT or sterile insect technique in the management of *B. dorsalis* in real time.

Similar to our observations, the sperms of *C. capitata* were present inside the females for more than 42 days [48]. Although field studies revealed that some *C. capitata* females had no sperms in the spermathecae, sperms existed in the ventral receptacle [29]. This indicates that observing sperm presence in both the spermathecae and ventral receptacle can more accurately determine the PMF. However, our results indicate that sperms are more readily and accurately observed in the spermathecae than in the ventral receptacle (Figure 5a and Figure 7a).

It has been reported that the maturity stages of *B. dorsalis* females were readily assessed by the ovarian index (length × width) [25]. However, can the changes in ovarian size determine the mating status of *B. dorsalis* females? Our results indicate that the superficial area of the ovary and spermatheca remains constant between mated and virgin females (Figure 4). This suggests that the observation of ovarian and spermathecal sizes is not applicable as a method to determine the mating status of *B. dorsalis* females.

It has been reported that after a pest management program of 4 months using MAT in the papaya orchard, the number of *B. dorsalis* males decreased by nearly 99%, but the fruit infestation rate only decreased by 48% [13]. Our laboratory results indicated that when the sex ratio of the population was 100:1, compared with 100:100, the PMF still reached 2% despite the number of males decreasing by nearly 99% (Figure 8). This indicates that although the PMF significantly reduces with the application of MAT, the control efficiency in the field is limited. Given the high dispersal capacity of *B. dorsalis* [49], one possible reason is that mated females from surrounding habitats migrate into the experimental area, leading to only a slight reduction in the fruit infestation rate. Another possible reason is the continuous migration of males from outside into the experimental area, which prevents a significant reduction in the PMF. Hence, it is evident that continuously monitoring the changes in PMF of *B. dorsalis* females in the field is vital for the scientific application of MAT. It is noteworthy that one possible explanation is that even a few mated females can cause considerable fruit damage, which needs to be confirmed by combining the PMF with fruit infestation rates.

Whether from the perspective of pest population monitoring or field management approaches, developing the *B. dorsalis* female lure technique is crucial. On the one hand, the application of MAT within the areawide pest management program remains an indispensable approach to managing *B. dorsalis* for the foreseeable future. On the other hand, the female lure technique may provide an optimal solution for the eco-friendly management of *B. dorsalis* [44]. To scientifically evaluate the effectiveness of MAT, using the female lure to monitor changes in the PMF in the field (including migratory females) is of significant value for timely adjustments of management strategies and technology applications, as demonstrated by the experimental results of Cunningham and Suda [13].

## 5. Conclusions

Sperms can be detected in the spermathecae and the ventral receptacle of mated *B. dorsalis* females. The sperms are more easily observed in the spermathecae and can be retained for more than 50 days. This indicates that dissecting the spermathecae to observe the presence of sperms is a reliable method to accurately monitor the PMF of *B. dorsalis* in the field. The method can be used in real time to accurately assess the control efficiency of MAT in *B. dorsalis* management. However, this research only verified the method’s reliability for determining the PMF in the field. We have not systematically studied the correlation between fruit infestation rate and the PMF in the field. Therefore, further research is needed to fully explore the PMF.

## Figures and Tables

**Figure 1 insects-15-00796-f001:**
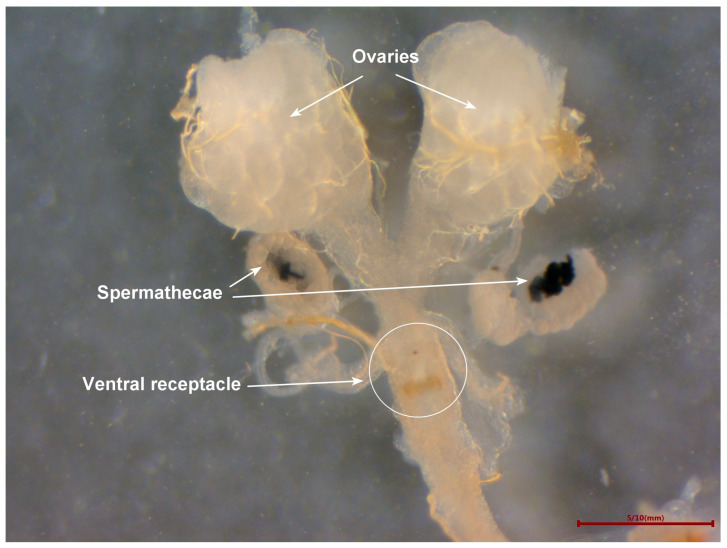
Reproductive tract of a *Bactrocera dorsalis* female showing the ovary, spermatheca, and ventral receptacle.

**Figure 2 insects-15-00796-f002:**
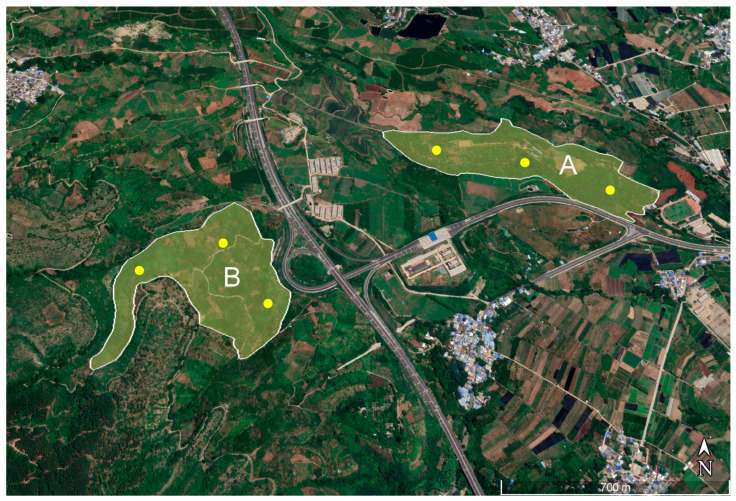
The geographic location of the field survey site in Yuanjiang County, Yuxi City, Yunnan Province, China. A indicates a wild mango orchard (untreated), and B indicates a wild mango orchard with the male annihilation technique (MAT) applied. Three trap points were established in each of the orchards, indicated by yellow dots.

**Figure 3 insects-15-00796-f003:**
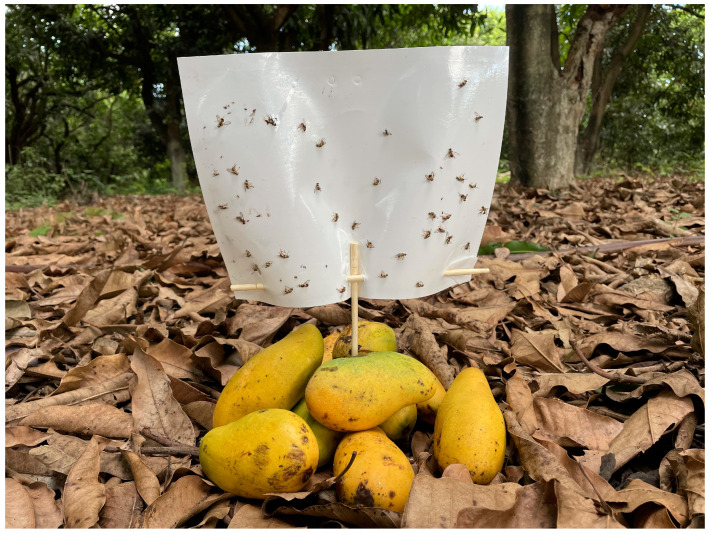
The method of using a combination of mangoes and sticky boards to trap *Bactrocera dorsalis* in the field experiment area, showing the effectiveness of the lure at 7 PM. Before setting up the trap points, all the mangoes were collected on the ground within the experimental area. Five mangoes with intact skin and five mangoes with splits (caused by falling) or wounds (insect bites on the skin) were selected and gathered into a pile (approximately 600 g). A sticky board was fixed above the pile.

**Figure 4 insects-15-00796-f004:**
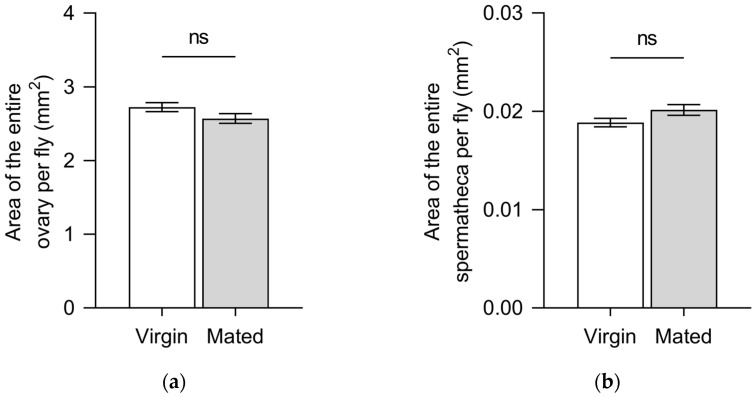
(**a**) Mean ± standard error (SE) ovarian superficial area of 10-day-old virgin *Bactrocera dorsalis* females and 10-day-old mated females (mated the previous day). (**b**) Mean ± SE superficial spermatheca area of 10-day-old virgin *B. dorsalis* females and 10-day-old mated females (mated the previous day). ns: not significant (*p* > 0.05).

**Figure 5 insects-15-00796-f005:**
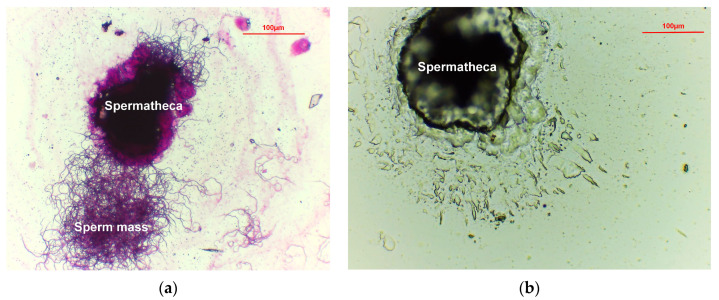
Torn spermatheca of a 10-day-old mated *Bactrocera dorsalis* female (**a**) showing sperm mass dyed with acid fuchsin. Torn spermatheca of a 10-day-old unmated *B. dorsalis* female (**b**).

**Figure 6 insects-15-00796-f006:**
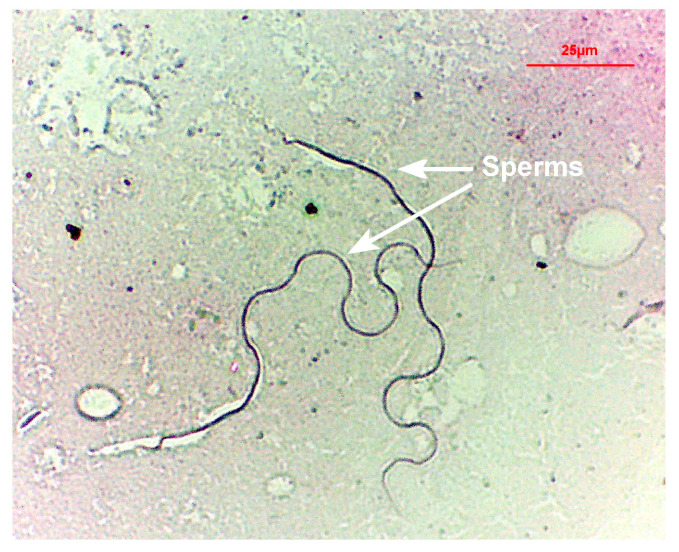
Sperms from torn spermatheca of a *Bactrocera dorsalis* female (dyed with acid fuchsin).

**Figure 7 insects-15-00796-f007:**
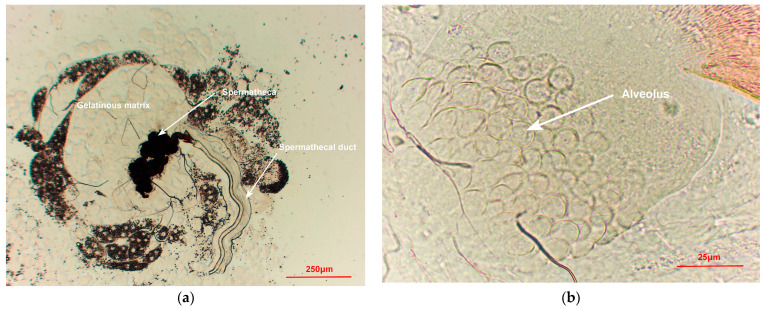
Pressed spermatheca of a *Bactrocera dorsalis* female (**a**) showing the spermatheca, spermathecal duct, and gelatinous matrix. The ventral receptacle of a *B. dorsalis* female (**b**) showing the alveolus.

**Figure 8 insects-15-00796-f008:**
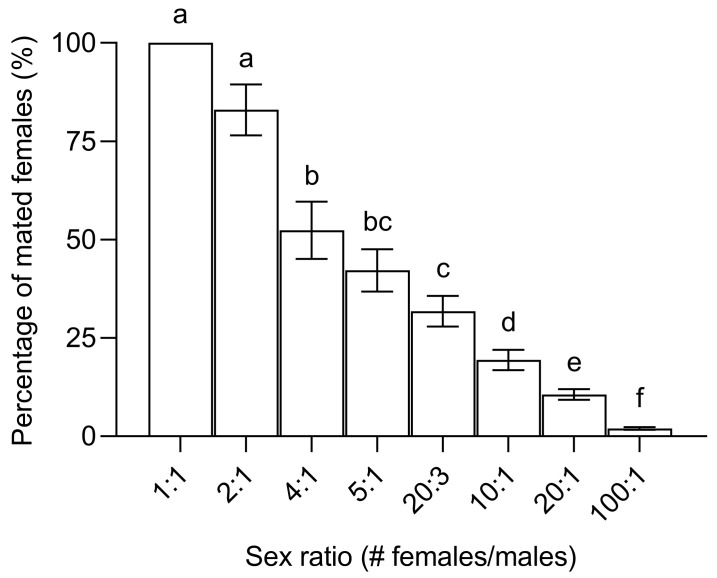
Mean ± SE proportion of mated *Bactrocera dorsalis* females in experimental cages with different sex ratios (number of females/males). Bars topped by different lowercase letters indicate significant differences (*p* < 0.05) according to Tukey’s honestly significant difference (HSD) test.

**Figure 9 insects-15-00796-f009:**
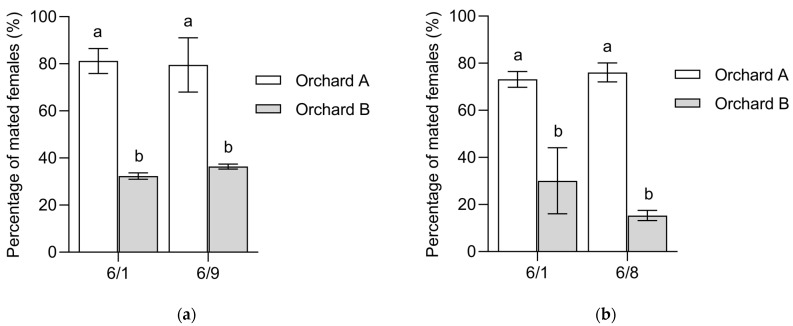
(**a**) Mean ± SE proportion of mated *Bactrocera dorsalis* females in two orchards on 1 June (Orchard A, *n* = 41; Orchard B, *n* = 49) and 9 June (Orchard A, *n* = 38; Orchard B, *n* = 60), 2022. (**b**) Mean ± SE proportion of mated *B. dorsalis* females in two orchards on June 1 (Orchard A, *n* = 43; Orchard B, *n* = 30) and 8 June (Orchard A, *n* = 52; Orchard B, *n* = 38), 2023. Bars topped by different lowercase letters indicate significant differences (*p* < 0.05) according to Tukey’s honestly significant difference (HSD) test.

## Data Availability

The data presented in this study are available on request from the corresponding author.

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
