# Peer review of "Evaluating the Efficacy of the Male Annihilation Technique in Managing Oriental Fruit Fly (Diptera: Tephritidae) Populations through Microscopic Assessment of Female Spermathecae"

_insects, 2024, doi:10.3390/insects15100796_

Round 1
Reviewer 1 Report
Comments and Suggestions for Authors
The manuscript "Evaluating the efficacy of male annihilation technique in managing Oriental fruit fly (Diptera: Tephritidae) populations through microscopic assessment of female spermathecae" provides a detailed way to determine the mating rates of Bactrocera dorsalis females in the field to assess the efficacy of the male annihilation technique (MAT). The topic is interesting enough to investigate and the experiments are well-designed, though somehow incomplete in the one performed in the field (see below). Accurate evaluation of a control measure is of paramount relevance for IPM strategies.
I would recommend its acceptance for publication once the authors consider the comments provided here and in the pdf file.
First the authors should acknowledge that the end point of any control measure is the reduction of fruit damage or infestation level. To this end, a way to evaluate this is well established. Yet, identifying the correlation with fruit infestation level and proportion of females with sperm in the spermathecae in MAT treated orchards as compared to control plots will surely contribute to better understand reasons of failure or areas of improvement.
In this sense, this study provides relevant information and procedures required to determine mating rate in the field. Yet, it fails to correlate it with fruit infestation levels, which could have made the study much significant. I recommend that the authors discuss this a bit further.
In general, the text is well written. However, the discussion section repeats the results; these should be avoided. Additionally, though I am not native in English, I can see it can be improved. A proof reding will surely contribute.
Additional comments are provided in the attached file.

Though I am not native in English, I can see it can be improved. A proof reding will surely contribute.
Author Response
Thank you very much for taking the time to review this manuscript. Please see the attachment.

Reviewer 2 Report
Comments and Suggestions for Authors
A sound and unambiguous study. My main criticism is the use of the term 'mating rate' throughout the manuscript. Mating rate implies the number of copulations in a given time period (eg once per day or twice per week etc). I believe that in recording presence/absence of sperm in spermathecae, you are determining mating status (or fertility status). Thus, by way of example, on line 19 you write "...female mating rate was less than 81.2%...". It is more accurate to say something along the lines of '...81% of females had sperm in their spermathecae..'.
Further line 22 - presence of sperm is NOT an accurate monitor of mating rate. It is an accurate indicator of mating status. Thus, I suggest the removal of mating rate throughout and replace with mating status (or something similar).
Lines 217 to 225: I felt the descriptions and stats contained within this text was adequately captured in Fig 8. Maybe replace with a more generic comment that indicates that as the SR became more female biased, the % of females with sperm in their spermathecae decreased (and then refer to Fig 8).
Line 235 - please say how many females from each group were assayed for sperm presence.
Line 306 - what about female migration?
Line 329 - what exactly do you mean by multiple experimental points?
Comments on the Quality of English LanguageGood
Author Response

(The authors gave the same response as above.)

Reviewer 3 Report
Comments and Suggestions for Authors
In the introduction part the authors should add part regarding field sterility assessment for other fruit flies, especially C. capitata that is similar methodology as for the B. dorsalis.
MM part is well described in details, only Fig. 1 fits to the results part. No further comments.
Results and conclusions: well argumented, no further comments
Author Response

(The authors gave the same response as above.)
